# Ion Channels Involved in Substance P-Mediated Nociception and Antinociception

**DOI:** 10.3390/ijms20071596

**Published:** 2019-03-30

**Authors:** Chu-Ting Chang, Bo-Yang Jiang, Chih-Cheng Chen

**Affiliations:** 1Institute of Biomedical Sciences, Academia Sinica, Taipei 115, Taiwan; chuting82@gmail.com (C.-T.C.); qwert985432@gmail.com (B.-Y.J.); 2Taiwan Mouse Clinic, National Comprehensive Mouse Phenotyping and Drug Testing Center, Taipei 115, Taiwan

**Keywords:** substance P, NK1R, nociception, anti-nociception, pain

## Abstract

Substance P (SP), an 11-amino-acid neuropeptide, has long been considered an effector of pain. However, accumulating studies have proposed a paradoxical role of SP in anti-nociception. Here, we review studies of SP-mediated nociception and anti-nociception in terms of peptide features, SP-modulated ion channels, and differential effector systems underlying neurokinin 1 receptors (NK1Rs) in differential cell types to elucidate the effect of SP and further our understanding of SP in anti-nociception. Most importantly, understanding the anti-nociceptive SP-NK1R pathway would provide new insights for analgesic drug development.

## 1. Background

Substance P (SP) was first described by von Euler and Gaddum, in 1931 [1]. The authors observed an unknown substance that stimulated contraction of the intestine ex vivo. This substance was identified and Euler and Gaddum named it substance P, from the bottle containing it, labeled P1, P2 etc., meaning “powder”. In the 1970s, SP was homogeneously purified by Chang and Leeman [2] and was later determined to be an 11 amino-acid peptide, H-Arg Pro Lys Pro Gln Gln Phe Phe Gly Leu Met-NH_2_ (RPKPQQFFGLM), with an amidation at the C-terminus [3]. SP belongs to the tachykinin family and serves as a neurotransmitter and a neuromodulator. It is encoded by preprotachykinin-1 (or tachykinin 1 (*TAC1*)) that produces SP and neurokinin A via alternative slicing and post-translational modifications [4].

SP is widely distributed in the human body, especially in nervous systems and inflammatory cells. A general assumption of the SP action describes SP as a neuropeptide released from pain-sensing fibers (nociceptors) to increase pain sensitivity through its actions in the dorsal horn of the spinal cord [5]. SP also triggers proinflammatory cytokine release resulting in inflammation, vasodilation and plasma extravasation [6]. Although considerable evidence indicates that SP transmits pain signaling and serves as a mediator of pain [7,8,9], accumulating studies reveal that SP also has an anti-nociceptive effect [10,11]. Interestingly, evidence showing the antinociceptive role of SP has been built over several years and can be dated back to 1976 [10]. The discrepant function of SP is believed to depend on the cell type regarding the expression of neurokinin receptors with unique underlying effector systems modulating differential ion channels.

## 2. SP-Mediated Signaling

There are three tachykinin receptors: Neurokinin 1 (NK1), 2 (NK2), and 3 (NK3). SP preferentially binds to NK1 receptor (NK1R). NKRs are G-protein-coupled receptors located in both the central nervous system (CNS) and peripheral nervous system (PNS). G proteins have Gα, Gβ, and Gγ subunits. The Gα subunit is classified into 5 families (G_s_, G_i_, G_o_, G_q/11_, and G_12/13_). In most cases, NK1R is coupled to the pertussis toxin-insensitive G_q/11_ cascade [12,13]; nevertheless, crosstalk with other G proteins such as G_12/13_ [14], G_o_, and G_s_ [15] has been reported. In G_q/11_-mediated signaling, phospholipase C is activated by G_βγ_ binding, which in turn results in hydrolysis of membrane phospholipid PtdIns(4,5)P2 to form diacylglycerol (DAG) and inositol triphosphate (IP3). IP3 further triggers calcium release in cytoplasm from sarcoplasmic reticulum, whereas DAG activates protein kinase C (PKC) and results in calcium influx from the L-type calcium channel in the plasma membrane [16]. The increase in calcium ion in the cytoplasm further leads to an array of cell responses.

There are two natural forms of NK1R: A full-length receptor with 407 amino acids and a truncated receptor with 311 amino acids, lacking 96 amino acids in the carboxyl terminal [17,18]. The human *NK1R* gene contains five exons. The truncated version is generated when the intron between exons 4 and 5 is not removed that encounters a premature stop codon before the start of exon 5. These two types of NK1Rs have differential features. The binding affinity of SP is 10 times less to the truncated NK1R than the full-length form. Furthermore, the underlying signaling of the two different NK1Rs differs. The carboxyl terminus of full-length NK1R is a crucial element for G-protein coupling. The truncated form of NK1R impairs the ability of G-protein binding and results in G-protein-independent mechanisms. Indeed, the truncated form of NK1R fails to interact with β-arrestin, an important protein mediating the desensitization and internalization of activated G-protein-coupled receptors [19]. Furthermore, activation of the truncated NK1R has different effects on calcium mobilization, phosphorylation of PKCδ, extracellular signal-regulated kinase 1/2, and regulation of interleukin 8 mRNA expression as compared with the full-length NK1R [20].

The two NK1Rs are differentially distributed in peripheral tissues and in the nervous system. In the brain, full-length form is abundantly expressed in striatum, caudate nucleus, putamen, globus pallidus, nucleus accumbent, and hypothalamus; whereas the truncated NK1R expression is relatively low in the brain, and most represented in the PNS and peripheral tissues including heart, lung, prostate, and bone [21]. To sum up, the different NK1R subtypes could trigger differential effector systems and further result in distinct cellular responses in different tissues and organs.

## 3. SP and Pain

Pain is an unpleasant sensation and can be divided into two major categories: Acute and chronic pain. Acute pain lasts only short time and can be ameliorated over time. Acute pain helps us avoid the physical damage to our body and serves as a warning signal. In contrast, chronic pain has no biological function and is a disease itself rather than symptom of a disease. Chronic pain usually lasts for more than six months. People with chronic pain represent about 10.1% to 55.2% of the population in various countries according to epidemiological study [22]. People with chronic pain often show other symptoms, such as fatigue, sleep disorder, memory problems, anxiety, and depression, which greatly affect their quality of life.

Chronic pain can be further divided into two subtypes: Inflammatory and neuropathic pain. Inflammatory pain is related to tissue injuries, which lead to inflammatory reactions. In contrast, neuropathic pain results from nerve injury and neuronal sensitization in the CNS and PNS. However, inflammatory pain and neuropathic pain are also closely correlated. Nerve injuries result in tissue inflammatory responses, which lead to inflammatory pain. Likewise, tissue inflammation triggers an inflammatory cascade releasing a variety of inflammatory mediators such as SP, calcitonin gene-related peptide, and neurokinin A to damage nerves [23,24]. Clinical studies have demonstrated that many forms of chronic pain have mixed components of inflammation and neuropathy [25].

Substantial evidence suggests that SP is the key element in neurogenic inflammation and has an important role in eliciting pain sensation in both the CNS and PNS. In the CNS, SP results in central sensitization by activating excitatory post-synaptic potential [26]. In the PNS, releasing SP from peripheral nociceptive nerve fibers can cause neurogenic inflammation in the skin [27,28]. In addition, people with fibromyalgia, a chronic pain disorder, showed elevated SP level in cerebrospinal fluid [29,30]. Chemical ablation of neurons expressing SP receptors in lamina I [31] and genetic disruption of the encoding gene of SP [32] or its receptor [33] reduced pain responses. Furthermore, a recent study showed that selectively ablating *TAC1*-expressing neurons in the spinal cord abolished the sustained pain but not the reflexive defensive response [34]. Together, these studies support that SP is an important signal molecule in pain transmission.

Thus, many studies have focused on developing selective NK1R antagonists as a potential analgesic drug. Although several preclinical studies showed an anti-nociceptive effect of NK1R antagonists, most clinical trials failed to show analgesia effects. The reasons for failure of most NK1R antagonists in clinical trials are still elusive. Several reasons for discrepancy between preclinical and clinical results have been proposed such as species differences in NKRs distribution [35], species differences in affinities to antagonists for NK1R [36,37,38], and the ability of animal models in predicting clinical pain [39]. The discovery of an antinociceptive effect of SP may also explain in part the failure of those clinical trials [10]. One hypothesis is that neurons innervating distinct locations may respond differently to SP—some neurons excited by SP and some inhibited by SP. Thus, elucidation of SP-mediated responses in different tissues and organs becomes a crucial step in developing specific NK1 antagonists as analgesic drugs.

## 4. SP-Mediated Anti-Nociception

The antinociceptive effect of SP was first reported in 1976 [10]. Although SP had been identified for more than four decades, in the early time, the impure natural SP contaminated by bradykinin or some other kinin-like compounds impeded SP research. In the 1970s, SP was homogenously synthesized. With purely synthetic SP, the effect of SP could be clearly deciphered. Stewart et al. first reported that SP treatment by intracerebral and intraperitoneal injection could produce naloxone-reversible analgesia, and the site of action was in the CNS [9]. Subsequently, several studies confirmed the SP-mediated anti-nociception via opioid receptors in the CNS, and SP seemed to be a regulatory peptide to normalize the responses to pain stimuli. In 1978, Frederickson et al. claimed that a small amount of SP (1.25 to 5 ng per mouse) by intracerebroventricular injection produced a naloxone-reversed anti-nociception effect in mice [32]. However, higher doses (>50 ng per mouse) caused hyperalgesia. The authors also reported that although the C-terminus of SP (SP6–11) is very similar to that of endogenous opioid peptides, neither SP nor the SP6–11 acted on opioid receptors. At low doses, SP triggered the release of endorphins but at higher doses, directly excited neurons in the brain [40]. In 1980, Oehme et al. suggested that SP produced naloxone-reversed analgesia in mice with high sensitivity to thermal stimulation but induced hyperalgesia in mice with low sensitivity to thermal stimulation [41]. In addition, SP has been found to effectively reduce neuropathic pain [42] and inflammatory pain [43]. To sum up, these studies demonstrated that SP can regulates opioid-dependent analgesic effects in distinct cell types, probably via different receptors.

Because it has also been reported that the C-terminus of SP is sufficient for biological activities in nociception [44], a certain active fragment of SP may be essential for anti-nociception. Indeed, evidence has shown that the N-terminal and C-terminal domains of SP have opposite functions. Hall and Stewart demonstrated that the N-terminus of SP (SP1–7) was related to naloxone-reversible anti-nociceptive and anti-aggressive actions, whereas the C-terminus (SP7-11) was thought to mediate pain transmission [45,46]. Furthermore, Skilling et al. showed that N-terminus of SP (SP1–7) but not the C-terminus (SP5–11) inhibited the release of excitatory neurotransmitters into spinal-cord extracellular fluid, which was reversed by naloxone [47]. These studies agree with the findings of the naloxone-reversible analgesic effect by SP in the CNS and PNS. The 11 amino-acid SP may be cleaved by enzymatic degradation into differential fragments. Those fragments could interact with differential receptors to induce an anti-nociceptive effect possibly via release of met-enkephalin or other endogenous opioid peptides [48,49,50].

Other evidence has shown that SP could act on NK1R to modulate opioid receptors [51,52]. Bowman and colleagues showed that SP increased the recycling of mu-opioid receptors in sensory neurons and led to elevated sensitivity of opioids [53]. Together, these studies suggest that the anti-nociceptive role of SP could act via opioid signaling.

## 5. SP-Mediated Anti-Nociception in Muscle

Accumulating evidence has shown a role for SP in anti-nociception in the PNS, especially in muscle. Despite much evidence indicating that SP can cause cutaneous pain, applying SP to muscle induced neither neurogenic inflammation nor painful perception in humans and rats [54,55,56]. In contrast, Lin et al. showed that SP had an anti-nociceptive role in muscle rather than causing pain [57]. With whole-cell patch clamp recordings on dissociated muscle-afferent dorsal root ganglion (DRG) neurons, the authors revealed that SP attenuated the acid sensing ion channel 3 (ASIC3)-induced inward current by enhancing M-channel-like potassium current. ASIC3 is a voltage-independent sodium channel activated by the extracellular protons. It has been found as a molecular determinant involved in pain-associated tissue acidosis [58,59]. As well, a recent study showed that ASIC3 can detect extracellular acidification and also respond to mechanical stimuli [60,61].

The in vivo antinociceptive role of SP was demonstrated in a rodent model of chronic widespread muscle pain induced by dual intramuscular acid injections, one of the fibromyalgia pain models developed by Sluka et al., in 2001 [62]. Two injections, separated by one to five days, of pH-4 acidic saline in the unilateral gastrocnemius muscle in rodents produced chronic and bilateral mechanical hyperalgesia of hind paws and muscle that required activation of ASIC3 [63]. Blocking ASIC3 activation, at the first or the second or both acid injections, abolished the induction and development of chronic muscle hyperalgesia. Furthermore, in the dual acid-injection model, the first acid injection could depolarize ASIC3-expressing muscle nociceptors and also simultaneously trigger SP release, which further enhanced the M-channel-like potassium current to attenuate ASIC3-induced depolarization in gastrocnemius muscle-afferent DRG neurons. In mice lacking *TAC1* (no SP and neurokinin A production), chronic pain could be induced by a single acid injection, which suggests that the anti-nociceptive effect was produced by the first acid injection but was diminished with the second injection [57]. The reason for ineffective SP in a second acid injection is still unclear and requires further investigation.

Regarding the acid-induced anti-nociception via SP release, the other important question is what types of acid sensors contribute to the release of SP as an anti-nociceptive acid sensor. The anti-nociceptive acid sensors are still unknown. Although previous study indicated that an acid sensor other than ASIC3 and transient receptor potential cation channel subfamily V member 1 (TRPV1) could trigger SP release [64], the possibility of the co-contribution of ASIC3, TRPV1, and other acid sensors such as other ASIC subtypes and/or proton-sensing G protein-coupled receptors is still not excluded. A recent study demonstrated that low-level laser therapy (LLLT) was effective in reducing mechanical hyperalgesia in the dual acid-injection model. The analgesic mechanism is associated with activation of TRPV1 to release SP in muscle [65]. This study provides new insights regarding the involvement of TRPV1 in acid-mediated anti-nociception. Furthermore, it reveals the involvement of SP in LLLT analgesia, which is widely used in pain control for musculoskeletal pain in the field of physical medicine and rehabilitation. In light of the antinociceptive role of SP in muscle, NK1R agonists might be promising candidates for pain relief in intractable musculoskeletal pain, such as fibromyalgia.

## 6. Ion Channels Involved in SP Signaling

SP can modulate a variety of ion channels (Table 1) resulting in an increase or decrease of neuronal excitability [66]. In most studies, SP excites neurons by increasing the function of excitatory ion channels and decreasing that of inhibitory ion channels. For example, SP has been shown to excite neurons by elevating the conductance of sodium channels and decreasing that of potassium channels in locus coeruleus neurons [67]. SP also inhibits inwardly rectifying K^+^ channels in nucleus basalis neurons via G_q/11_ [68,69], and inhibits Ca^2+^-activated potassium channels [I_K(Ca)_] in stellate ganglion neurons via pertussis toxin-insensitive G proteins [70]. Other studies showed that SP can inhibit the N-type calcium channel in sympathetic neurons via pertussis toxin-insensitive G proteins [71,72]. The above studies suggest that SP mainly modulates ion channel activity via the G-protein-dependent pathway. However, non-G-protein effector systems are also reported in SP-mediated signaling. Lu and colleagues revealed that SP-induced increase of sodium conductance was mediated by activating the sodium ion-permeable cation channel complex of NALCN (sodium leak channel, non-selective) and UNC-80 in mouse hippocampal and ventral tegmental area neurons independent of G-protein but mediated by Src family tyrosine kinase [73]. Accordingly, SP can modulate diverse channels and activate the neurons by G-protein-dependent or -independent signaling.

A few studies showed that SP hyperpolarizes neurons in the PNS. SP hyperpolarized vagal sensory neurons of ferrets by inducing a Ca^2+^-dependent outward potassium current [74]. SP decreased non-selective cation channel conductance in outer hair cells of guinea pig cochlea [75]. SP enhanced the M-type potassium current independent of G-protein but dependent on tyrosine kinase in half of muscle-afferent DRG neurons [57]. Similarly, an SP-mediated G_i/o_-dependent pathway could augment the M-type potassium current in DRG neurons and trigeminal ganglion (TG) neurons [76]. Finally, SP could inhibit T-type calcium channels in DRG and TG neurons [77]. Together, SP can modulate a variety of ion channels via different signaling pathways, which are cell type-specific.

## 7. Ion Channels Involved in SP-Mediated Nociception

SP has a well-known role in the transmission of nociceptive information in the spinal cord. In the supraspinal level, microinjection of SP in the rostral ventromedial medulla (RVM) has also been found to induce hyperalgesia via descending facilitation mechanisms in a glutamate- and GABA_A_-dependent manner [92,93]. Furthermore, SP release is increased in the spinal dorsal horn after peripheral nociceptive stimulation [94]. Overall, the amount of nociceptive neurotransmitter released by primary afferent nerve terminals determines the level of pain. The release of SP in the spinal dorsal horn would interact with glutamate to enhance peripheral inputs. Previously, SP and glutamate were found to co-exist in small-diameter DRG neurons and their nerve terminals in the spinal dorsal horn [95]. Glutamate acts as the molecule transmitting the fast excitatory signal, whereas SP modulates relatively slow excitatory synapse responses [96]. Besides, SP signaling can enhance the *N*-methyl-*D*-aspartate (NMDA) channel function leading to greater pain sensitivity [78,79,80,81]. Thus, the two transmitters, glutamate and SP, are considered to interact and convey the nociceptive information in the spinal cord.

In the peripheral system, SP is an important element in neurogenic inflammation causing extravasation and sensory neuron sensitization. During inflammatory processes, inflammatory cells and peripheral nerve terminals release SP, which, in turn, modulates a variety of ion channels rendering sensitization of sensory neurons in an autocrine or paracrine manner. In the PNS, SP mainly exists in the small sensory nociceptors. Release of SP can act on NK1R via differential intracellular mechanisms to potentiate the channel activities of TRPV1 [84,85,86], Nav1.8 [83], and l- and N-type calcium channels [87] in a subset of small-diameter DRG neurons, thereby resulting in hyperalgesia. SP could also decease the activity of low-threshold potassium channel (kv4) in capsaicin-sensitive DRG neurons and thus sensitize the nociceptors [88]. In the orofacial region, SP can potentiate the P2X3 receptor in TG neurons, leading to elevated pain sensitivity [89]. In summary, SP predominantly acts on peripheral sensory small neurons (presumably nociceptors) to excite the neurons, thereby increasing nociceptive responses.

## 8. Ion Channels Involved in SP-Mediated Anti-Nociception

The role of SP in anti-nociception has been confirmed in both the CNS and the PNS. In the CNS, the analgesic effect of SP is mainly associated with opioid-dependent pathways, although other studies also demonstrated the involvement of GABA_A_R and glycine receptors in the lamina V of the spinal cord [90,97].

In the PNS, SP can act on NK1R via either tyrosine kinase or G_i/o_ effector system to potentiate the inhibitory M-type potassium channels and inhibit excitatory T-type calcium channels in a specific subset of sensory neurons [76,77]. In muscle afferent DRG neurons, the SP-NK1R signaling is coupled with tyrosine kinase to enhance M-type potassium channel activity [57]. Enhancing the M-type potassium current and inhibiting the T-type calcium current results in dampening neuronal excitability, in turn further leading to an anti-nociceptive effect (Figure 1).

## 9. M-type Potassium Channels

M-type potassium channels are voltage-gated potassium channels (M for muscarine) encoded by *KCNQ* genes. *KCNQ* genes encode five Kv7 subunits (Kv7.1–7.5) [98]. Kv7.2, 7.3, 7.4, and 7.5 are expressed in the nervous system. The M-type potassium channel mainly features Kv7.2 and Kv7.3, although other subunits can also contribute to the formation of M channels in some locations.

M channels were first discovered in sympathetic neurons [99]. When muscarinic agonists activate the muscarinic acetylcholine receptors, sympathetic neurons become more responsive to the synaptic inputs and can become burst firing, rather than fire a single spike. This situation is due to the suppression of a unique channel, which led to the name “M” channel.

M-type channels are able to conduct a non-inactivating outward current with a threshold of about −60 to −80 mV and regulated by many neurotransmitters such as SP and bradykinin. The biophysical features of this current include slow activation and deactivation potassium current. Because M channels can be opened near the resting membrane potential, they also play a role in clamping the resting membrane potential, so they are ideally suited to control neuronal excitability.

M channels are functionally expressed in various central and peripheral neurons including hippocampal [100] and DRG neurons [101]. *KCNQ2*, *3*, and *5* are variably expressed in sensory neurons, including small-diameter nociceptors and large-diameter proprioceptors. Because pathological pain such as inflammatory and neuropathic pain features for neuronal hypersensitivity responds to nociceptive inputs, M-type potassium channels are considered a potential analgesic target in controlling nociceptive excitability. Opening of M channels results in hyperpolarization of the neurons, which decreases cell membrane excitability. Accordingly, the M-channel openers retigabine and flupirtine can effectively attenuate muscle pain [102] and inflammatory pain [101].

In 2012, Lin and colleagues showed that SP-enhanced M channel activity in muscle-afferent DRG neurons via an unconventional signal pathway by activating NK1R coupled with phosphotyrosine kinase to attenuate the mechanical hyperalgesia [57]. Then, Gamper’s group demonstrated that SP could augment the M channel in a subset DRG neurons mainly by acting on NK1R via Gi/o and a redox-dependent pathway [76]. The similar result but discrepant cellular mechanisms could be due to the different subset of DRG neurons examined. The SP-enhanced M current via the phosphotyrosine kinase pathway is prominent on medium to large DRG neurons expressing ASIC3 innervating muscle, whereas the SP-enhanced M current via the Gi/o pathway acts mainly on small neurons expressing TRPV1 innervating skin.

These studies provide insight into SP-mediated anti-nociception in local tissues. Activating the SP-NK1R pathway specifically in muscle afferents becomes an attractive therapeutic target to treat chronic pain. Although the pro-nociceptive effect of SP in the spinal cord has been well documented, local peripheral application of SP can be diluted and has little effect on the spinal cord. Thus, targeting the SP-NK1R pathway in the PNS creates a strategy for pain relief without producing severe side effects in the CNS.

## 10. T-type Calcium Channels

T-type calcium channels (Cav3) are low-voltage activated calcium channels opened with small membrane depolarization and can generate Ca^2+^-dependent burst firing, pacemaker activity, and low-amplitude neuronal oscillation [103]. In contrast to high-threshold L-type calcium channels (long-lasting calcium channels), T-type calcium channels only mediate a “transient” calcium influx, with the voltage threshold about −60 with slower activation. T channels also form a window current in the resting membrane potential controlling the sub-threshold neuronal excitability.

The α1 subunit consists of the major channel pore of T-type calcium channels, allowing for calcium influx through it. The channel pore-forming α subunits include Cav3.1, Cav3.2, and Cav3.3 encoded by calcium voltage-gated channel subunit alpha1 G (CACNA1G), H, and I, respectively. T-type calcium channels were first found in small sensory neurons by Carbone and Lux in 1984 [104]. Subsequently, functional expression of T-type channels was confirmed in nociceptive DRG neurons, with Cav3.2 as the predominant isoform. Nelson et al. further characterized the expression profile of T channels and indicated that a subtype of DRG neurons highly expressed T channels, named T-rich cells, which were also highly sensitive to capsaicin and ATP stimuli [105]. Those T-rich cells are mainly small-diameter neurons with narrower APs and after depolarizing potentials during the action potential falling phase. Altering gating parameters or directly increasing the amplitude of the T current can result in burst firing and neuronal excitability [105]. Inhibiting the T-type channels can effectively reduce the neuropathic and inflammatory pain [106,107,108].

Recently, Huang et al. demonstrated that T channels were modulated by SP [77]. The authors found that SP release in damaging tissue would act on NK1R via Gi/o to release mitochondria reactive oxygen species, which further inhibited the T channels. This study revealed SP-mediated anti-nociception in a subtype of small-diameter sensory neurons. Further studies are required to determine whether medium- to large-diameter sensory neurons also show the same modulation of SP in T-type calcium channels.

## 11. Conclusions

Accumulating evidence has shown SP-mediated anti-nociception in both the CNS and PNS. However, SP has a mixed effect of pro-nociception and anti-nociception in the nervous system. Especially, blocking NK1R signaling in the CNS can result in adverse outcomes. In contrast, the anti-nociceptive effect of SP in the PNS has been well characterized, especially in muscle. Applying SP to muscle-afferent DRG neurons enhances the M-channel outward current, which further hyperpolarizes the neurons. Accordingly, locally targeting the SP-NK1R pathway on muscle afferent neurons could be a promising analgesic strategy.

## Figures and Tables

**Figure 1 ijms-20-01596-f001:**
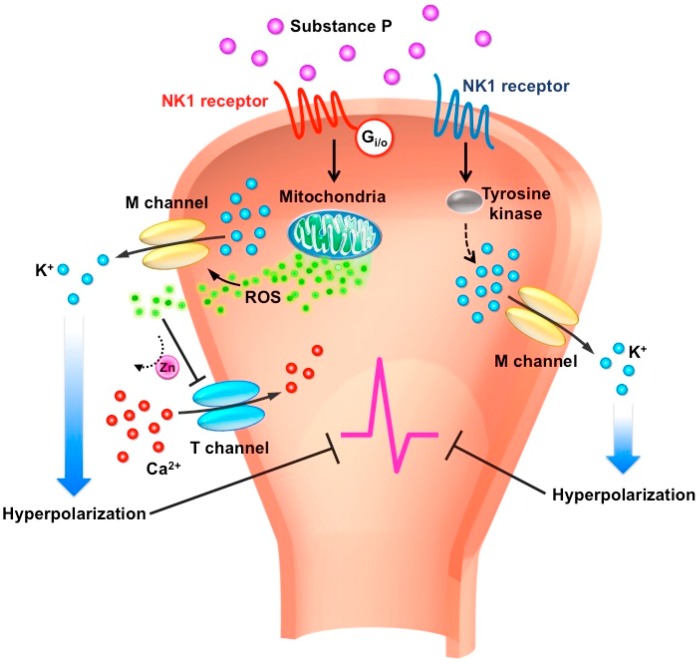
Schematic diagram of substance P-mediated signaling and ion channels in the peripheral sensory neurons. Release of substance P (SP) in the nerve terminal acts on neurokinin 1 receptor (NK1R) via two different effector systems modulating M-type K^+^ and T-type Ca^2+^ channels. First, activated NK1R coupled to tyrosine kinase augments the M-type potassium channels, resulting in neuronal hyperpolarization. Second, activated NK1R coupled to G_i/o_ triggers reactive oxygen species (ROS) release from mitochondria simultaneously to augment M-type potassium channels and inhibit T-type calcium channels, which inhibits neural firing in peripheral sensory neurons.

**Table 1 ijms-20-01596-t001:** Ion channels modulated by substance P.

Channel Types	Effector System	Cell Types	Species	Effects on Current	Outcomes	References
NMDAR	PKC	Spinal dorsal horn neurons, spinal thalamic neurons	Rat, monkey	↑	Pro-nociception	[78,79,80,81]
K_ir_	G_q/11_ and PLC-β1	Locus coeruleus neurons, nucleus basalis neurons	Rat	↓		[67,68,69]
K_Ca_	PTX-insensitive G protein	Stellate ganglion neurons	Guinea pig	↓		[70]
N-type Ca^2+^ channel	PTX-insensitive G protein	Stellate ganglion neurons, superior cervical ganglion neurons, sympathetic neurons	Guinea pig, rat, frog	↓		[70,71,72]
NALCN	Src family kinases	Hippocampal and ventral tegmental area neurons	Mice	↑		[73]
TRP3C	Via NK2R	HEK293		↑		[82]
Nav1.8	PKCε	DRG	Rat	↑	Pro-nociception	[83]
TRPV1	PKCε	DRG	Rat	↑	Pro-nociception	[84,85,86]
L, N type calcium channel	PKC	DRG	Rat	↑		[87]
Low threshold potassium channel (Kv4)		DRG	Rat	↓	Pro-nociception	[88]
P2X3		TG	Rat	↑	Pro-nociception	[89]
GABA_A_R	G_i/o_	Spinal dorsal horn neurons	Rat	↑	Anti-nociception	[90]
Glycine receptor	G_i/o_	Spinal dorsal horn neurons	Rat	↑	Anti-nociception	[90]
M-type potassium channel	Tyrosine kinase	DRG	Mice	↑	Anti-nociception	[57]
M-type potassium channel	G_i/o_	DRG, TG	Rat	↑	Anti-nociception	[76]
T-type calcium channel	G_i/o_	DRG	Rat	↓	Anti-nociception	[77]
K_Ca_		Vagal sensory neurons	Ferret	↑		[74]
I_h_		Vagal sensory neurons	Ferret	↓		[91]
Non-selective cation channel	PTX-insensitive G protein	Outer hair cells of cochlea	Guinea pig	↓		[75]

Abbreviations: NMDAR, *N*-methyl-*D*-aspartate receptor; K_ir_, inward rectifier potassium channel; K_Ca_, calcium-activated potassium channel; I_h_, hyperpolarization-activated channel; PLC, phospholipase C; PTX, pertussis toxin; PKC, Ca^2+^/phospholipid-dependent protein kinase; DRG, dorsal root ganglion; TG, trigeminal ganglion; Symbols: ↑, increase; ↓, decrease.

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
