# Peer review of "Ion Channels Involved in Substance P-Mediated Nociception and Antinociception"

_ijms, 2019, doi:10.3390/ijms20071596_

Round 1

Reviewer 1 Report

This is an interesting review on the anti-nociceptive role of SP and ion channels involved in SP-mediated nociceptive and anti-nociceptive mediation. Generally, the whole manuscript is very well prepared and authors had provided adequate information on the field.

Some minor correction;

1.     NK1R should be reported same over the whole text, e, g; page 2 lines 61, 92, 02

2.     Can authors introduce abbreviations in table 1

3.     In Table 1, column with “effects on current” is empty, can it be removed from the table 1?

Author Response

Point-to-point responses:

Reviewer #1:

1. NK1R should be reported same over the whole text, e, g; page 2 lines 61, 92, 02

Thank you for your comment. I already correct them as NK1R.

2. Can authors introduce abbreviations in table 1

As suggested, we have added the abbreviations below the table 1.

3. In Table 1, column with “effects on current” is empty, can it be removed from the table 1?

Thanks! In the column with “effect on current”, we used arrows as symbols to indicate the effect of SP channel-mediated current (increase or decrease). It might be lost during the file conversion to PDF. We have made sure it is good in the revised version.

Reviewer 2 Report

 The submitted review by Chang et al., covers the topic of substance P-induced effects on ion channels and how this can bidirectionally regulate nociception.  Overall, the review is well written and well organized.  Specific comments for revision are listed below:

-In my experience, SP is typically described as a neuropeptide that increases pain sensitivity through its actions in the dorsal horn of the spinal cord. How controversial is it to suggest that SP can also have anti-nociceptive effects?  The authors do a good job of describing studies where antinociceptive effects were observed.  However, they could do a bit better at putting this into the context of general assumptions about SP actions.  Is SP-mediated anti-nociception a recently observed phenomenon, or has this evidence been building over several years?

-Page 2, lines 59-62: Please provide a bit more information about where specifically these receptors are distributed.

-The first two paragraphs of section 3 require more references to support the information presented.

-Page 2, line 94: Elaborate upon clinical trial effects.  There are only two references here.  Have there only been two clinical trials performed? Also, there are a number of reasons that could account for discrepancies between preclinical and clinical data that should be discussed including the specific compounds used, receptor occupancy achieved at the doses delivered, pharmacogenetic interactions, etc.  Are there any species differences in receptor distribution or affinity that could explain incongruous observations in preclinical versus clinical findings?

-Does SP act in higher brain regions to influence pain sensation, or are its actions in relation to nociception specifically localized to the periphery and spinal cord?

-For the non-traditional effects of SP, is this occurring via the NK1R but through different intracellular mechanisms, or is SP ever directly interacting with a different receptor or the ion channel itself.  It is generally made clear that non-Gq related mechanisms are taking place in some preparations, but it should be made more clear in some areas of the manuscript what exactly SP is binding to.  The figure clarifies some of these relationships (NK1R coupling to novel intracellular messengers), but it is unclear if it is suggested that SP is binding to ion channels in some cases.

-In section 11, it would be good to briefly discuss some specific clinical conditions that could be targeted by influencing activity of the SP-NK1R system in peripheral muscle. 

Author Response

Point-to-point responses:

Reviewer #2:

Comments: In my experience, SP is typically described as a neuropeptide that increases pain sensitivity through its actions in the dorsal horn of the spinal cord. How controversial is it to suggest that SP can also have anti-nociceptive effects?  The authors do a good job of describing studies where antinociceptive effects were observed.  However, they could do a bit better at putting this into the context of general assumptions about SP actions.  Is SP-mediated anti-nociception a recently observed phenomenon, or has this evidence been building over several years?

Responses: Thanks for the comments. The discovery of SP in antinociception can be dated back to 1976 and the history is described in the section— SP-mediated antinociception. General assumption of SP is its role in enhancing pain. However, mounting evidence demonstrated SP is able to induced both nociceptive and antinociceptive effect in both CNS and PNS. Therefore, we discuss here several features of SP and SP- modulated differential cellular signaling and ion channels in hope of elucidating the effect of SP. As suggested, we have added some discussion to highlight the SP history in the second paragraph of the Background section.

Comments: -Page 2, lines 59-62: Please provide a bit more information about where specifically these receptors are distributed.

Responses: Thank you for your comment. We have added some detail expression profile of the full-length and truncated forms of NK1R in lines 70-75.

Comments: -The first two paragraphs of section 3 require more references to support the information presented.

Responses: Thank you for your comment. We have added some references regarding the epidemiology study of chronic pain and the description of inflammatory and neuropathic pain part.

Comments: -Page 2, line 94: Elaborate upon clinical trial effects.  There are only two references here.  Have there only been two clinical trials performed? Also, there are a number of reasons that could account for discrepancies between preclinical and clinical data that should be discussed including the specific compounds used, receptor occupancy achieved at the doses delivered, pharmacogenetic interactions, etc.  Are there any species differences in receptor distribution or affinity that could explain incongruous observations in preclinical versus clinical findings?

Responses: Thank you for your comment. Indeed, there are several possible reasons for failure of NKIR antagonist in clinical trials for analgesia, in terms of species differences in neurokinin receptor distribution, species differences in affinities to NK1R antagonists. Moreover, some papers even discuss whether the applied animal models are predictive or not for the NK1R antagonists. We have added the above discussion with references in lines 111-115.

Comments: -Does SP act in higher brain regions to influence pain sensation, or are its actions in relation to nociception specifically localized to the periphery and spinal cord?

Responses: We have added the effect of SP in modulating descending facilitation mechanisms to enhance pain sensitivity in the supraspinal cord level in lines 242-245.

Comments: -For the non-traditional effects of SP, is this occurring via the NK1R but through different intracellular mechanisms, or is SP ever directly interacting with a different receptor or the ion channel itself. It is generally made clear that non-Gq related mechanisms are taking place in some preparations, but it should be made more clear in some areas of the manuscript what exactly SP is binding to.  The figure clarifies some of these relationships (NK1R coupling to novel intracellular messengers), but it is unclear if it is suggested that SP is binding to ion channels in some cases.

Responses: Thank you for your comment. SP does act on NK1R coupled with different intracellular mechanisms. We have discussed more details of the SP-NK1R signaling in lines 276-280. 

Comments: In section 11, it would be good to briefly discuss some specific clinical conditions that could be targeted by influencing activity of the SP-NK1R system in peripheral muscle.

Responses: Thanks! Our newly published paper (in Pain Medicine) has shown that Low-level laser therapy can relieve pain via SP release in a mouse model of fibromyalgia. We have discussed more about the clinical application of the SP antinociception in lines 199-203.
